# Identification of Potential Inflammation-Related Genes and Key Pathways Associated with Complex Regional Pain Syndrome

**DOI:** 10.3390/biom13050772

**Published:** 2023-04-29

**Authors:** He Zhu, Bei Wen, Li Xu, Yuguang Huang

**Affiliations:** Department of Anesthesiology, Peking Union Medical College Hospital, Chinese Academy of Medical Sciences, Peking Union Medical College, Beijing 100730, China

**Keywords:** complex regional pain syndrome, hub genes, protein–protein interaction, inflammatory response

## Abstract

Complex regional pain syndrome (CRPS) is a chronic pain that affects the extremities after a trauma or nerve injury with no definite established treatment. The mechanisms mediating CRPS are not completely elucidated. Thus, we conducted a bioinformatics analysis to identify hub genes and key pathways to determine strategies for more effective treatments of CRPS. Finally, there is only one expression profile of GSE47063 in terms of homo sapiens-based CRPS from the Gene Expression Omnibus (GEO) database, which included four patients and five controls. We explored the differentially expressed genes (DEGs) in the dataset and conducted Gene Ontology (GO) functional and Kyoto Encyclopedia of Genes and Genomes (KEGG) enrichment analysis of the potential hub genes. A protein–protein interaction (PPI) network was also established; then, according to the score of each hub genes, we used R software to draw the nomogram model to predict the rate of CRPS. Furthermore, GSEA analysis was estimated and assessed by the normalized enrichment score (NES). From the GO and KEGG analysis, we identified the top five hub genes (MMP9, PTGS2, CXCL8, OSM, TLN1); all of the selected DEGs were mainly enriched in their inflammatory response. In addition, the GSEA analysis showed complement and coagulation cascades also play an important role in CRPS. This study, to our knowledge, is the first to conduct further PPI network and GSEA analyses. Thus, targeting excessive inflammation could offer new therapeutic methods for CRPS and related physical and psychiatric disorders.

## 1. Introduction

Complex regional pain syndrome (CRPS) is a chronic and disabling pain, with a prevalence of approximately 5.4–26.2 per 100,000 person years, with females being affected 3–4 times more than males [1]. A longitudinal study including 1549 patients with fracture of carpal bones, the distal radius or ulna or both, showed a 3.8% incidence of CRPS [2]. There are some risk factors that have been identified currently: female gender; age over 70 years; upper limb involvement; and intraarticular, dislocated and high-energy fractures; however, the underlying pathophysiology seems to be complex and controversial; some potential mechanisms have been suggested, including cytokine imbalance and neuroinflammation [3,4]. Moreover, CRPS has a large impact on daily life, with people commonly presenting with allodynia, hyperalgesia, skin temperature changes, and oedema, which all bring a heavy financial burden to families [5].

At the moment, the attention we pay to CRPS is not enough; it is distinct from other pain syndromes by the presence of autonomic dysfunction and persistent regional inflammatory changes, with a lack of dermatomal distribution; the clinical diagnosis of CRPS was established using the “Budapest criteria”, published by the International Association for the Study of Pain (IASP) 2007 [6]. Recently, CRPS has been further divided into two subtypes, based on the absence (CRPS I, previously known as reflex sympathetic dystrophy) or presence (CRPS II, previously known as causalgia) of a major nerve injury [5]. Although rat models (chronic post ischemia pain or tibial fracture/casting rodent model) provide valuable evidence for different aspects of the pathophysiology of acute posttraumatic pain and inflammation, these models do not produce all the long-term symptoms that patients experience in some way [7].

The discovery of effective therapeutic targets for CRPS is vitally imperative because of the lack of effective treatments. Increased evidence suggests that inflammation, as characterized by an activation of glial cells and infiltration of leukocytes, has a critical role in the pathogenesis of CRPS [8]. Moreover, inflammatory processes have also been postulated to contribute to specific syndromes such as depression and other behavioral manifestations common to this syndrome in CRPS (sleep disorders, decreased activity, decreased social interaction). A growing body of evidence suggests that inflammatory mediators, including proinflammatory cytokines, chemokines, and growth factors in dorsal root ganglion and the spinal cord, contribute substantially to the genesis and maintenance of CRPS. Previous research found that HLA-A29.1, MMP9, ANPEP, HDC, G-CSF3R, and STAT3 were associated with CRPS patients, but they did not conduct further PPI network and GSEA analysis [9].

Microarray is a mature high-throughput method for genome-wide gene expression profiling. Bioinformatics can provide tools for the analysis of large amounts of information from the GEO database to identify hub genes and key pathways for new clinical treatment strategies [10]. Therefore, this study aimed to discuss the molecular mechanisms and pathogenesis of CRPS, thus providing a new foundation for the prevention of this life-altering condition.

## 2. Materials and Methods

### 2.1. Microarray Datasets of CRPS

The gene expression microarray GSE47063 (9 samples: 2 CRPS I, 2 CRPS II, and 5 controls) was downloaded from the Gene Expression Omnibus (GEO) database (http://www.ncbi.nlm.nih.gov/geo, accessed on 5 November 2022), a subdataset of the National Center of Biotechnology Information (NCBI), which includes biological expression data of many species obtained via array, SNP array, and high-throughput sequencing [11]. All datasets in the present study were downloaded from public databases. These public databases allowed researchers to analyze for scientific purposes, and thus ethics approval was not required.

### 2.2. Differential Expression Analysis

We downloaded the gene expression series matrix, platform information, and clinical information from the GEO database. Then, the corresponding platform information was used to convert the probe IDs in the expression series matrix into gene symbols to obtain an international general expression series matrix with the row name of gene symbols and the column name of sample names. The mean value of multiple probe sets was calculated if they corresponded to the same gene. Next, we further used the tool of GEO2R (http://www.ncbi.nlm.nih.gov/geo/geo2r/, accessed on 5 November 2022) to analyze the differentially expressed genes (DEGs) in the CRPS group and the Control group. Results are presented in the browser as a table of the top 250 genes ranked by *p*-value, for RNA-seq; the table is the result of the Wald test when comparing 2 groups of samples. Highlighted genes are significantly differentially expressed at a default adjusted *p*-value cutoff of 0.05 (red = upregulated, blue = downregulated). The DEGs were defined as genes with a *p*-value < 0.05 and a |log (fold change (FC))| ≥ 1.

### 2.3. Gene Ontology and Pathway Enrichment Analysis

The gene enrichment analysis of the genes in the meaningful module was performed composed of Gene Ontology (GO) and Kyoto Encyclopedia of Genes and Genomes (KEGG) pathway enrichment through the ClusterProfiler package using R Software (version 4.0.3). Afterward, the enrichment results were visualized with the R package “enrichplot” to further analyze the biological functions and pathways. At the same time, Metascape (https://metascape.org/, accessed on 7 November 2022) combines functional enrichment, interactome analysis, gene annotation, and membership search for experimental biologists to comprehensively analyze and interpret gene expression profiling [12].

### 2.4. Protein–Protein Interaction (PPI) Network Analysis

The STRING database (STRING: functional protein association networks (string-db.org, accessed on 8 November 2022)) is an online search for known and predicted protein interactions [13]. At present, it stores at least 14,094 kinds of organisms, 67.6 million kinds of proteins, and more than 20 billion interaction data. We uploaded 37 DEGs to the string database and visualized a PPI network map of the String database using Cytoscape software. In Cytoscape, each gene was considered as a node and experimentally determined interactions were used as edge attributes. The important modules were identified and visualized using Molecular Complex Detection (MCODE), which is a plugin in Cytoscape for discovering densely connected nodes in a given network. We further used the MCODE module to calculate the core interactions in the PPI network and the Clugo to further analyze the key pathways.

### 2.5. Gene Set Enrichment Analyses (GSEA)

In brief, GSEA (http://software.broadinstitute.org/gsea/index.jsp, accessed on 8 November 2022) is routinely used to analyze and interpret coordinate pathway-level changes in transcriptomics experiments with less than seven samples per condition. Since its initial application to microarray experiments, GSEA has demonstrated utility across many applications, including RNA-seq gene expression experiments, genome-wide associations studies, and proteomics and metabolomics studies. To further study the potential pathways associated with CRPS for biological processes enrichment, GSEA enrichments were estimated using the normalized enrichment score (NES). The significance of the enrichment was assessed at an FDR < 0.25 level and *p*-value < 0.05.

### 2.6. Establishment of Predictive Models

First of all, we use cytoHubba in Cytoscape to screen out the top hub genes. In order to simplify the predictive model, we converted the expression score of each hub gene into binary variables through logistic regression. To further determine the convenience for clinical application, in our study, the DynNom package of R software was used to draw the nomogram to preliminarily predict the rate of CRPS. All statistical analyses were performed using the R programming language and environment (http://www.r-project.org/, accessed on 9 November 2022).

### 2.7. Ethical Declaration

All of the data used in this study were obtained from public databases; thus, this study does not contain any intervention experiments associated with animals or humans.

## 3. Results

### 3.1. Identification of CRPS-Associated Genes

We downloaded series Metrix and corresponding platform information (GPL10558) and clinical information from the NCBI website which contains high-throughput gene expression data, chips, and microarrays under the accession number GSE47603. These nine samples (four in the CRPS group, five in the Control group) were selected for deep analysis in our study in accordance with our aims. To find the genes that were tightly correlated with CRPS, differential analysis was run for the genes in the GSE. DEGs of the dataset were found by using the GEO2R tool with the cutoff criteria of *p* < 0.05 and |log (fold change)| ≥ 1. The results revealed that a total of thirty-seven DEGs (thirty-three upregulated and four downregulated) were identified from the CRPS group versus the Control group (Figure 1); the details are shown in Table 1.

Next, according to the DEGs we have identified, we present a heatmap of the differentially expressed genes in the GEO dataset that colors the samples–groups (Figure 2).

### 3.2. Functional Annotation

We performed the GO functional annotation and KEGG pathway enrichment analysis on the 37 DEGs. For biological processes (BPs), these DEGs mainly involved antigen processing, T cell activation, the integrin-mediated signaling pathway, and inflammatory response. For the cellular component (CC), the DEGs were associated with the MHC class II protein complex, reticulum membrane, golgi transport vesicle membrane, endocytic vesicle membrane, and lysosomal membrane (Figure 3).

In addition, the GO molecular function (MF) analysis revealed that the DEGs were principally involved in the peptide antigen binding, integrin binding, and actin filament binding (Table 2).

Together, KEGG databases were also selected for preferred pathway sources and were used to explore the function of genes in biological systems. For simplicity, only the significant terms were recorded (Table 3).

Finally, we used online software (http://www.bioinformatics.com.cn/, accessed on 8 November 2022) and Metascape to perform further visual analysis. In fact, Metascape was initially designed to support biologists, as we observed most gene-list analysis tools were bioinformatician-oriented rather than biologist-oriented. It enables users to apply popular bioinformatics analyses to gene and protein lists in order to make effective data-driven gene prioritization decisions (Figure 4).

### 3.3. PPI Network Analysis and GSEA Analysis

We used the String online database to analyze DEGs and construct a PPI network. Homo sapiens were used as the organism for subsequent analysis. Network visualization in String was transferred to Cytoscape software to explore target modules and hub genes (Figure 5).

In order to further clarify the critical role of CRPS, we used the cytoHubba in Cytoscape to screen out the top hub genes, MMP9, PTGS2, CXCL8, OSM, and TLN1; additionally, we found that MYH9, ZYX, and ACTN1 also play an important role in CRPS (Figure 6).

To simplify the predictive model, we converted the expression score of each hub gene into binary variables; then, we used R software to draw the nomogram to predict the rate of CRPS, in consideration of the convenience for clinical application; the dynamic nomogram has been plotted using the DynNom package (Figure 7).

At the same time, GSEA employs a competitive null hypothesis to test significance; thus, we screened out one commonly significant enriched pathway through GSEA analysis (Version 4.3.2): the system of complement and coagulation cascade cascades affected by inflammation (Figure 8). Collectively, the complement and coagulation system is a part of the innate immune system, comprising more than 50 soluble and membrane-bound proteins that mediate inflammatory responses and defend against microbes [15]. Complement activation leads to the generation of C3a, C5a, and the membrane attack complex (C5b-9), which causes a release of pro-inflammatory and procoagulant cytokines such as TNF and in IL-6 associated with CRPS. In our study, the power of GSEA lies in its use of gene sets, which provide a more stable and interpretable measure of biological functions and can show greater experimental and technical variation.

## 4. Discussion

The discovery of effective therapeutic targets for CRPS is vitally imperative because of the lack of effective treatments. There is currently little research on the pathogenesis of CRPS, and about 10% of cases develop without a clear clinical symptom [4]. A recent study divided CRPS into two categories: ‘warm’ and ‘cold’ CRPS. The warm subtype is characterized by a warm, red, edematous, and sweaty extremity, whereas the cold type is typically indicated by a cold, blue, less edematous extremity [16]. Previous studies have shown that many targeting inflammatory genes are closely related to CRPS [9]; however, with the continuous research into CRPS undertaken over the past 10 years, and the updated methods of bioinformation analysis, we hope that this research can benefit patients to some extent. Moreover, some new studies have found that acute inflammation and edema due to tissue damage caused ischemia reperfusion through abnormal pathways and changes in the nerves system, including bones and muscles [17]. An increasing amount of evidence suggests that disturbances of body representation and body perception are important features of the CRPS phenotype with the impaired two-point discrimination threshold, but a recent review clarified that this phenomenon also occurs in various nonneuropathic pain conditions [18,19].

In this study, we searched the GEO dataset and only found one expression profile of GSE47063 in relation to homo sapiens-based CRPS. According to a prospective cohort study, the independent factors that were associated with the CRPS type I included high-energy injuries, severe fractures, and female sex [20]. GO functional and KEGG enrichment analysis demonstrated that most overlapped DEGs were mainly enriched in their inflammatory response. From the functional perspective, inflammation in traumatic or nerve injury induces peripheral and central sensitization to limit further injury. As a result, we identified that the top five hub genes (MMP9, PTGS2, CXCL8, OSM, TLN1) and complement system were associated with CRPS, and thus constructed a protein PPI network.

A recent study on CRPS showed inflammatory and anti-inflammatory serum cytokines are potential biomarkers for CRPS, which is consistent with our results [7,21]. Strikingly, Lenz M et al. found that bilaterally proinflammatory TNF-α and MIP-1β were increased and anti-inflammatory IL-1RA was decreased in CRPS [22]. Morellini N et al. demonstrated that an interaction between cutaneous nerves and mast cells may contribute to CRPS, and loss of dermal nerve fibers might attenuate chemotactic signals [23]. Heyn J et al. [24] found that the decrease in Th17 is regulated by CD39+ Tregs and that the anti-inflammatory T-cell shift may be a mechanism of CRPS. Therefore, targeting the processes and molecules that are involved in inflammation could lead to better treatments for CRPS.

Converging lines of evidence suggest that matrix metalloproteinase MMP9 plays an important role in inflammation as well as in pain processes [25]. As expected, from our PPI analysis, we found MMP9, a secreted zinc metallopeptidase, was the most important gene in the development of CRPS. Zinc is an essential nutrient for human health and has anti-oxidative stress and anti-inflammatory functions, and we found zinc ion binding may be one of the important mechanisms of CRPS. Notably, a recent study found that MMP-2 and MMP-9 are differently expressed depending on the clinical phenotype in CRPS; ipsilateral MMP-2 and contralateral MMP-9 were lower in CRPS with trophic changes [26]. Such data may support a process by which contralateral pain progresses to non-injury sites in CRPS. Hossaini et al. also found that one-sided inflammation may increase pain systems in the contralateral cord, the c-fos activation pattern of spinal Gly/GABA neurons [27,28]. At the same time, after nerve injury, activated Schwann cells mediate the breakdown of the blood–nerve barrier via MMP-9, which promotes the recruitment of immune cells from the vasculature and their subsequent release of more pro-inflammatory mediators [29].

In the study of Zhou [30], they found PTGS2 is upregulated in atherosclerosis, which illustrated that the factor is involved in the early inflammatory response of blood vessels in some way. Chemokines, a special class of cytokines with more than 50 members, are a family of small, secreted proteins. In our analysis, we found that CXCL8 (known as IL-8) was an important protein-coding gene in CRPS. The small cytokine CXCL8 is known to be one of the most potent chemoattractant molecules; among several other functions, it is responsible for guiding neutrophils through the tissue matrix until they reach sites of injury [31,32]. According to Huang [33], activation of CXCL8 exacerbated inflammatory reactions in trophoblast cells by inducing TNF-α and IL-1β, which can result in unexplained recurrent pregnancy loss.

Oncostatin M (OSM) has been proven to be an inflammatory factor with multiple regulatory effects. Recent studies have demonstrated that the OSM signal pathway is an important approach for tumor–microglia interaction [34]. Interestingly, OSM is elevated in human obesity, and its specific receptor (OSMRβ) is also induced in conditions of obesity and insulin resistance [35,36]. Our PPI analysis found that type 1 diabetes, graft-versus-host disease, and autoimmune thyroid disease may be related to CRPS. Thus, for obese diabetes patients with CRPS, we should pay attention to the expression of OSM.

There are some studies that found that the human leukocyte antigen (HLA) system is associated with the pathophysiology of CRPS [37,38]. According to de Rooij [39], the involvement of HLA-B62 and HLA-DQ8 in CRPS with dystonia may indicate that these HLAs are implicated in the susceptibility or expression of the disease. Similarly, our study found that HLA-A, HLA-DQB1, and HLA-DRB1 play important roles in the occurrence and development of CRPS. Immunologic influences are also fundamental to CRPS; the increase in neuropeptides such as substance P and calcitonin gene-related peptide leads to the release of pro-inflammatory mediators such as TNF-a, IL-1b, and IL-6 [7,21]. Some researchers also found that aminopeptidase N (ANPEP), histidine decarboxylase (HDC), granulocyte colony-stimulating factor 3 receptor (G-CSF3R), Talin1 (TLN1), signal transducer, and the activator of transcription 3 (STAT3) gene expression, were significantly increased in CRPS patients, and were also all involved in signal transduction, cell motility, and immunity [9]. Previous studies have shown that the complement and coagulation cascades play key roles in innate immunity and are involved in the pathogenesis of inflammatory-related diseases [40]. Moreover, the complement system is composed of a large number of proteases that react with each other via proteolytic cleavage to induce inflammatory responses, which is consistent with our GSEA analysis. To the best of our knowledge, no prior studies have evaluated associations between the complement system and CRPS.

The treatment for CRPS is mainly directed toward physical and psychological therapy, neuropathic pain medications, anti-inflammatory medications, and sympathetic nerve blocking [41,42]. Many studies demonstrated that no good therapeutic results were achieved in the use of NSAIDs in CRPS [43,44]. Hyperbaric oxygen and spinal cord or dorsal root ganglion stimulation can also be used to treat CRPS [45,46,47]. Recently, Vitamin C was thought to be a form of intervention that acts by inhibiting pro-inflammatory pathways mediated through antioxidant mechanisms, but the recent meta-analysis showed no statistical significance in preventing CRPS with distal radial fractures [48,49]. At the same time, mesenchymal stem cells (MSCs) actively contribute to the microenvironment of injured tissues; given the plasticity of the immunoregulatory phenotype of MSCs, inflammatory status and concurrent use of immunosuppressants should be considered when administering MSCs for the treatment of inflammatory diseases [50]. Nevertheless, these divergent findings suggest that the long-term efficacy of the methods above is yet to be determined.

Our study also has some limitations. Firstly, our analysis was based on blood samples from homo sapiens and only included nine samples, since there are fewer clinical trials investigating CRPS, especially on the study of its mechanisms. Secondly, we did not conduct further experimental verification for ethical reasons and because of a lack of tissue samples. It is our hope that this bioinformatics analysis will promote further research on the interactions between central and peripheral inflammatory pathways that manifest in CRPS.

## 5. Conclusions

In summary, this study used integrated bioinformatics to detect these hub genes (MMP9, PTGS2, CXCL8, OSM, TLN1) and the complement system that are associated with CRPS. Obviously, a clearer understanding of the mechanism and pathological significance of the DEGs mainly enriched in their inflammatory response will provide new perspectives to design potential therapeutic targets to intervene in CRPS processes.

## Figures and Tables

**Figure 1 biomolecules-13-00772-f001:**
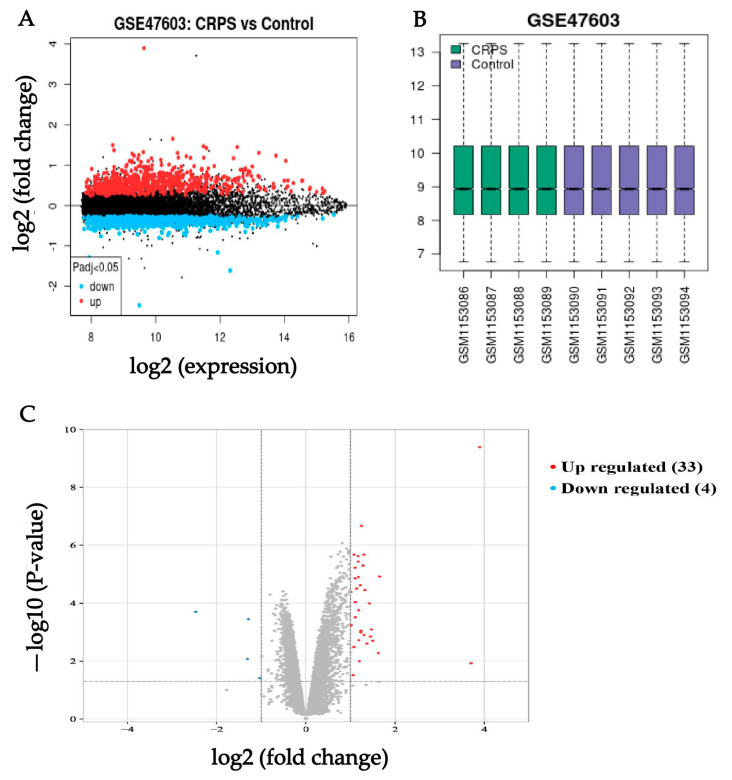
(**A**) Volcano of original gene expression from GEO dataset. (**B**) Box diagram of original gene expression from GEO dataset. (**C**) Identification of DEGs between the CRPS group and Control group in the GSE 47,063 dataset. Highlighted genes are significantly differentially expressed at a default *p*-value cutoff of 0.05 (red = upregulated, blue = downregulated).

**Figure 2 biomolecules-13-00772-f002:**
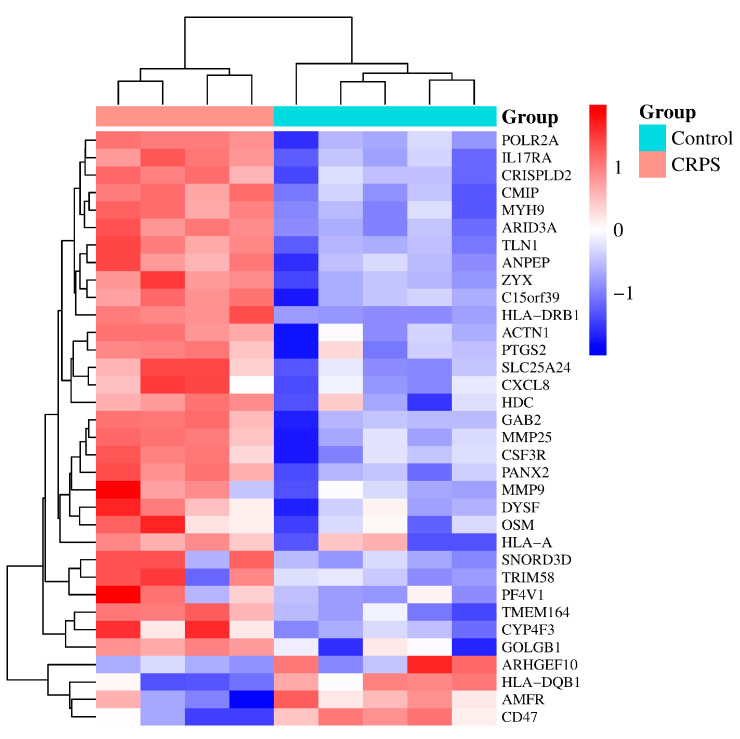
A heatmap based on gene expression dataset. *p* < 0.05 and |log (fold change)| ≥ 1. From Table 1 above, we found 3 duplicate genes; they are ZYX (ILMN_2371169, ILMN_1701875), CXCL8 (ILMN_2184373, ILMN_1666733), and PTGS2 (ILMN_1677511, ILMN_2054297). Some of them were automatically deleted when we drew the heatmap.

**Figure 3 biomolecules-13-00772-f003:**
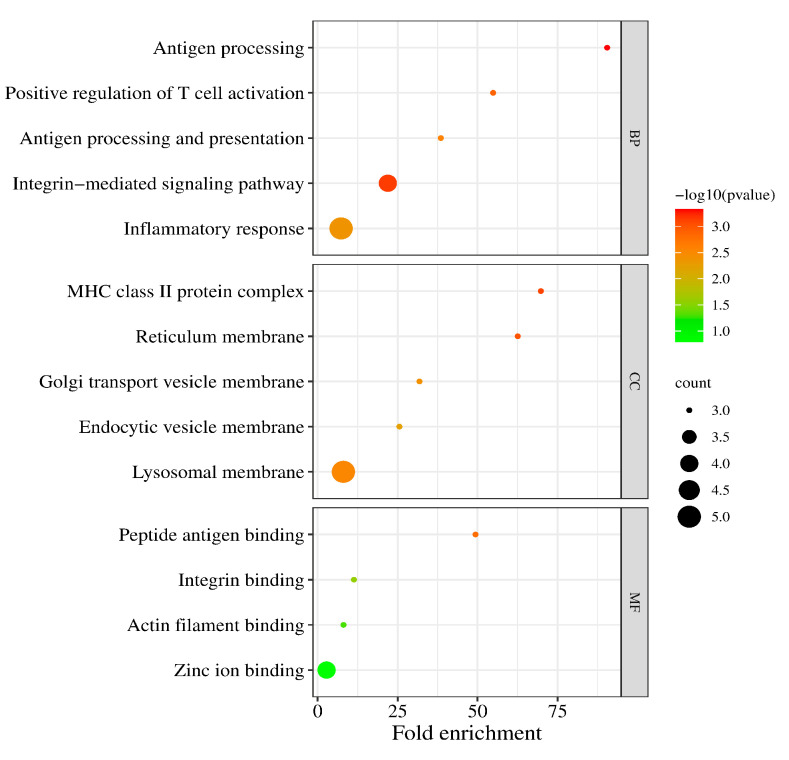
The enriched biological (BP), cellular component (CC), and molecular function (MF) terms in the GO analysis.

**Figure 4 biomolecules-13-00772-f004:**
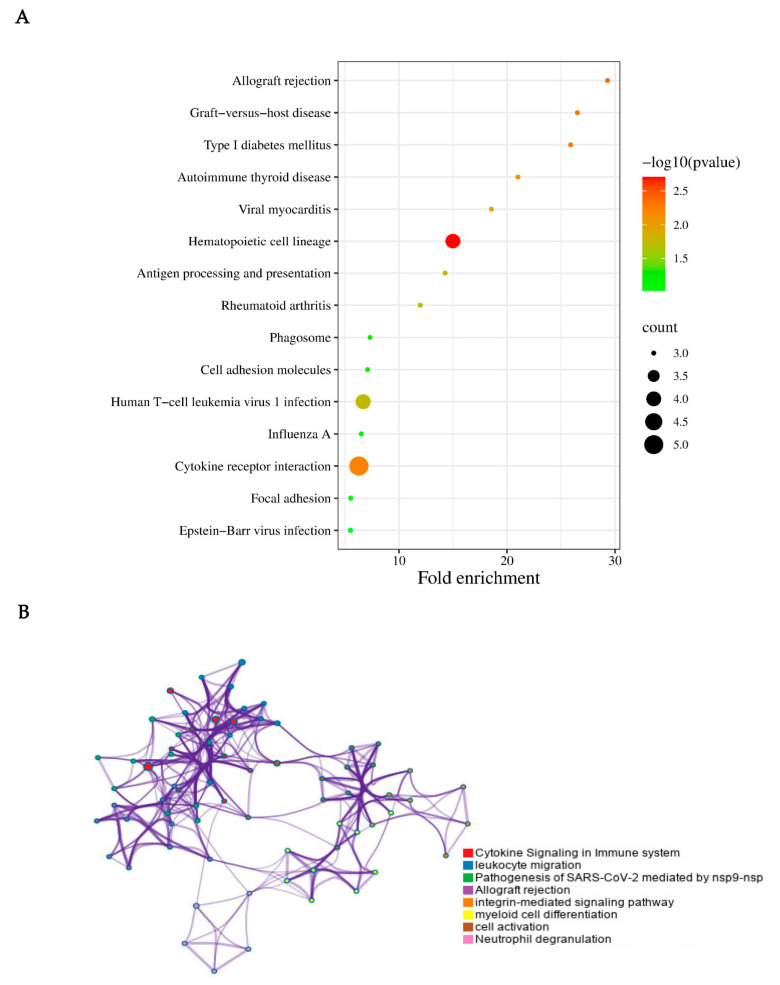
KEGG pathway enrichment analysis of the integrated DEGs. (**A**) KEGG pathway enrichment analysis on the 37 DEGs. The cytokine receptor interaction, human T-cell leukemia virus 1 infection, and hematopoietic cell lineage may play important roles in CRPS. (**B**) Visual analysis was performed using Metascape. In consideration of the result performed via Metascape, the cytokine signaling in the immune system, leukocyte migration, pathogenesis of SARS-CoV-2, allograft rejection, integrin-mediated signaling pathway, myeloid cell differentiation, cell activation, and neutrophil degranulation were closely associated with CRPS.

**Figure 5 biomolecules-13-00772-f005:**
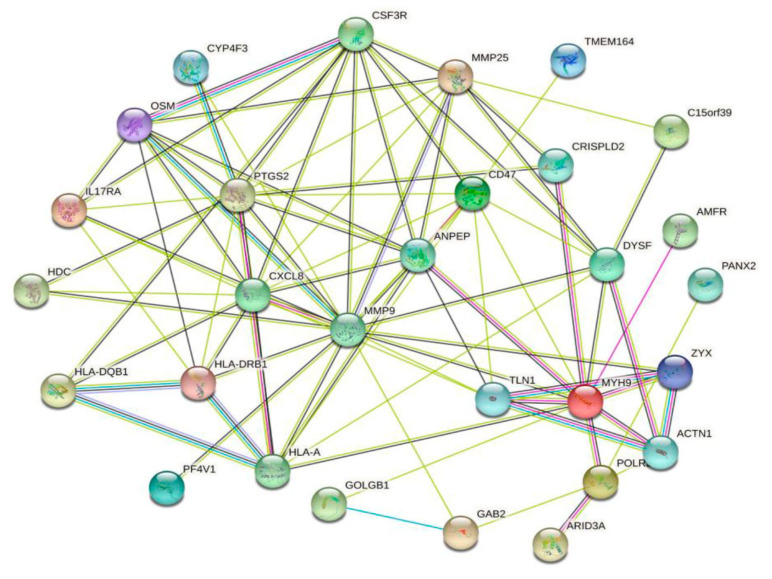
The PPI network of the DEGs, constructed using the Cytoscape software.

**Figure 6 biomolecules-13-00772-f006:**
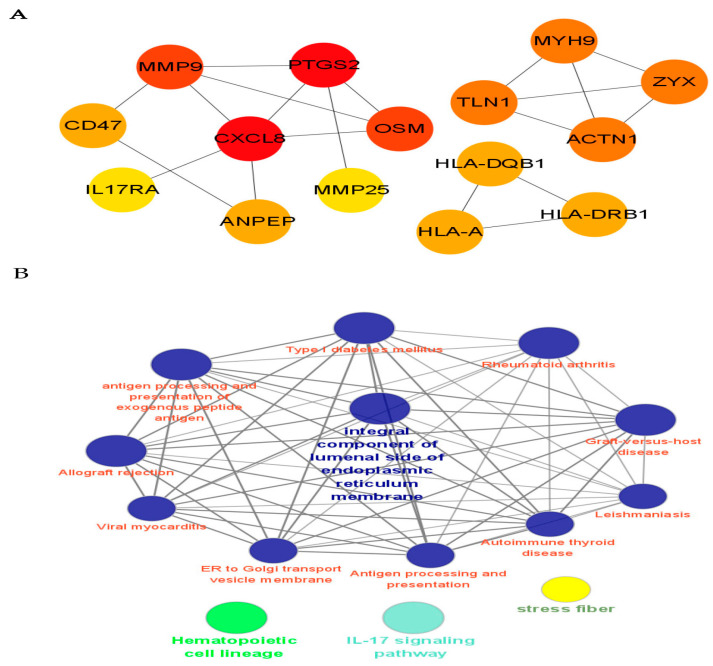
PPI network analysis and identification of hub genes. (**A**) We identified the top hub genes (MMP9, PTGS2, CXCL8, OSM, TLN1) associated with CRPS. (**B**) Cytoscape provides basic functionality to lay out and query the network associated with CRPS; to visually integrate the network with expression profiles, phenotypes, and other molecular states; and to link the network to databases of functional annotations [14]. The potential mechanism of CRPS is analyzed via the PPI network.

**Figure 7 biomolecules-13-00772-f007:**
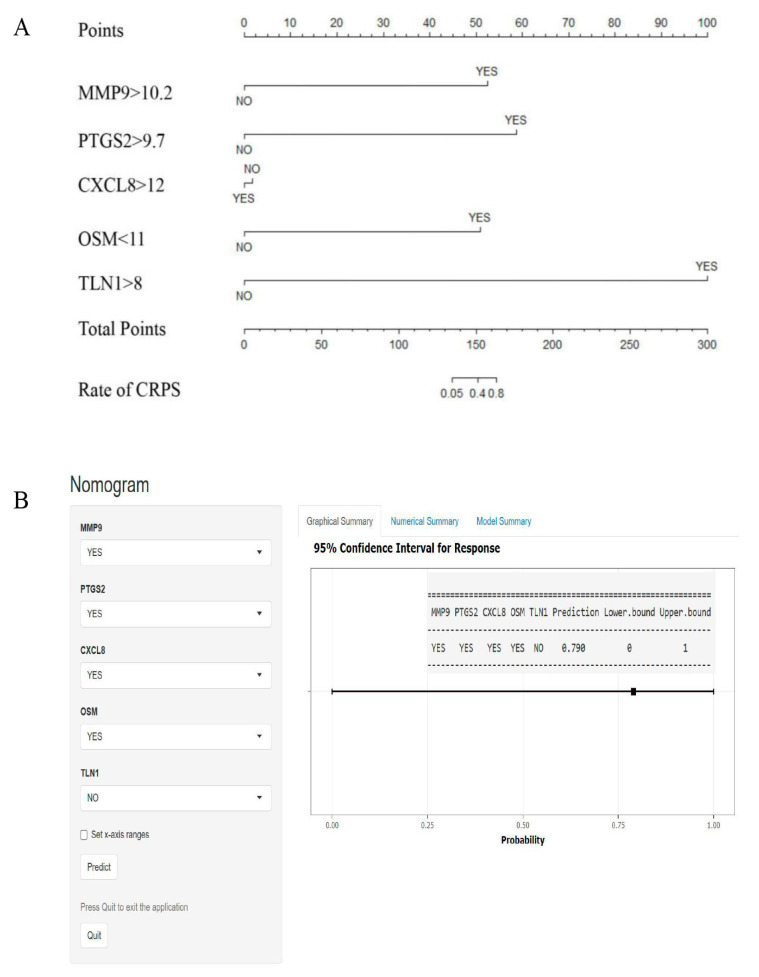
Nomogram for predicting CRPS. (**A**) According to the score of each independent variable, the vertical projection to the uppermost points score axis can correspond to a score, and finally all 5 independent risk factor variables are scored and counted. (**B**) Based on the hub genes, dynamic nomograms are plotted. For example, patients with scores for MMP9 > 10.2, PTGS2 > 9.7, CXCL8 > 12, OSM < 11, and TLN1 < 8; the prediction rate of CRPS is about 79%.

**Figure 8 biomolecules-13-00772-f008:**
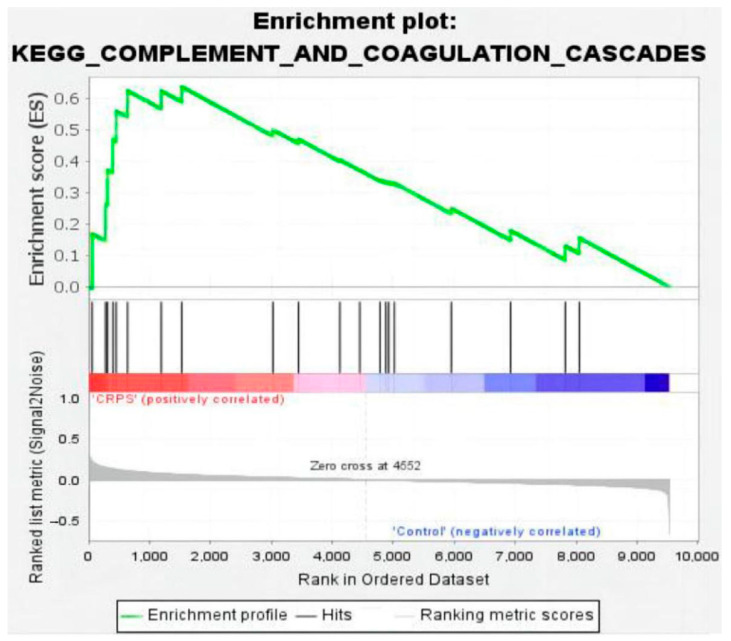
Enrichment plot for complement and coagulation cascades (nominal *p* value < 0.01, FDR < 0.25, NES = 1.88).

**Table 1 biomolecules-13-00772-t001:** DEGs of the dataset between the CRPS and Control groups.

ID	Gene Symbol	*p* Value	Log Fc	Gene Title
ILMN_1715169	HLA-DRB1	4.06 × 10^−10^	3.90	Major histocompatibility complex, class II, DR beta 1
ILMN_1670130	ARID3A	2.17 × 10^−7^	3.71	AT-rich interaction domain 3A
ILMN_1738075	CMIP	2.14 × 10^−6^	1.65	c-Maf inducing protein
ILMN_1696643	TLN1	2.14 × 10^−6^	1.63	Talin 1
ILMN_2371169	ZYX	2.42 × 10^−6^	1.50	Zyxin
ILMN_1722872	MYH9	3.71 × 10^−6^	1.47	Myosin, heavy chain 9, non-muscle
ILMN_1782385	POLR2A	5.04 × 10^−6^	1.45	RNA polymerase II subunit A
ILMN_1728724	IL17RA	5.91 × 10^−6^	1.42	Interleukin 17 receptor A
ILMN_1790689	CRISPLD2	1.19 × 10^−5^	1.37	Cysteine rich secretory protein LCCL domain containing 2
ILMN_1701875	ZYX	1.24 × 10^−5^	1.33	Zyxin
ILMN_1694810	PANX2	1.39 × 10^−5^	1.31	Pannexin 2
ILMN_3241091	TM × 10M164	2.38 × 10^−5^	1.30	Transmembrane protein 164
ILMN_1793729	C15orf39	3.11 × 10^−5^	1.28	Chromosome 15 open reading frame 39
ILMN_1763837	ANPEP	3.54 × 10^−5^	1.25	Alanyl aminopeptidase, membrane
ILMN_1665964	GAB2	4.09 × 10^−5^	1.24	GRB2 associated binding protein 2
ILMN_1711838	SLC25A24	9.17 × 10^−5^	1.23	Solute carrier family 25 member 24
ILMN_1717207	MMP25	1.01 × 10^−4^	1.22	Matrix metallopeptidase 25
ILMN_2232177	ACTN1	1.75 × 10^−4^	1.20	Actinin alpha 1
ILMN_1661266	HLA-DQB1	1.97 × 10^−4^	1.19	Major histocompatibility complex, class II, DQ beta 1
ILMN_2371280	CSF3R	3.06 × 10^−4^	1.18	Colony stimulating factor 3 receptor
ILMN_2356991	CD47	3.56 × 10^−4^	1.18	CD47 molecule
ILMN_1736190	CYP4F3	5.75 × 10^−4^	1.17	Cytochrome P450 family 4 subfamily F member 3
ILMN_1677511	PTGS2	8.13 × 10^−4^	1.17	Prostaglandin-endoperoxide synthase 2
ILMN_2184373	CXCL8	8.98 × 10^−4^	1.13	C-X-C motif chemokine ligand 8
ILMN_1792323	HDC	1.01 × 10^−3^	1.11	Histidine decarboxylase
ILMN_2054297	PTGS2	1.25 × 10^−3^	1.10	Prostaglandin-endoperoxide synthase 2
ILMN_1666733	CXCL8	1.43 × 10^−3^	1.10	C-X-C motif chemokine ligand 8
ILMN_1810420	DYSF	1.90 × 10^−3^	1.10	Dysferlin
ILMN_3242315	SNORD3D	1.98 × 10^−3^	1.08	Small nucleolar RNA, C/D box 3D
ILMN_1747935	GOLGB1	2.49 × 10^−3^	1.08	Olgin B1
ILMN_1780546	OSM	3.28 × 10^−3^	1.06	Oncostatin M
ILMN_1796316	MMP9	5.24 × 10^−3^	1.02	Matrix metallopeptidase 9
ILMN_1723116	AMFR	8.38 × 10^−3^	1.01	Autocrine motility factor receptor
ILMN_1745522	PF4V1	1.00 × 10^−2^	−1.04	Platelet factor 4 variant 1
ILMN_2165753	HLA-A	1.18 × 10^−2^	−1.29	Major histocompatibility complex, class I, A
ILMN_1705458	TRIM58	3.08 × 10^−2^	−1.31	Tripartite motif containing 58
ILMN_2132809	ARHGEF10	3.88 × 10^−2^	−2.47	Rho guanine nucleotide exchange factor 10

**Table 2 biomolecules-13-00772-t002:** GO terms and enrichment analysis of DEGs.

GO ID	Term	Count	*p* Value	Fold Enrichment
BP				
GO:0002504	Antigen processing	3	0.0005	90.5
GO:0050870	Positive regulation of T cell activation	3	0.0013	54.9
GO:0019882	Antigen processing and presentation	3	0.0026	38.5
GO:0007229	Integrin-mediated signaling pathway	4	0.0007	21.9
GO:0006954	Inflammatory response	5	0.0041	7.3
CC				
GO:0042613	MHC class II protein complex	3	0.0008	69.8
GO:0071556	Reticulum membrane	3	0.0010	62.6
GO:0012507	Golgi transport vesicle membrane	3	0.0038	31.8
GO:0030666	Endocytic vesicle membrane	3	0.0058	25.6
GO:0005765	Lysosomal membrane	5	0.0030	8.0
MF				
GO:0042605	Peptide antigen binding	3	0.0016	49.4
GO:0005178	Integrin binding	3	0.0269	11.3
GO:0051015	Actin filament binding	3	0.0499	8.1

Notes: BP, biological processes; CC, cellular component; MF, molecular function. The table shows GO terms. Enrichment analysis of DEGs with *p* < 0.05.

**Table 3 biomolecules-13-00772-t003:** KEGG pathway enrichment analysis of DEGs.

KEGG Pathway ID	Term	Count	*p* Value	Fold Enrichment
hsa05330	Allograft rejection	3	0.0042	29.3
hsa05332	Graft-versus-host disease	3	0.0051	26.5
hsa04940	Type I diabetes mellitus	3	0.0053	25.9
hsa05320	Autoimmune thyroid disease	3	0.0080	21.0
hsa05416	Viral myocarditis	3	0.0102	18.6
hsa04640	Hematopoietic cell lineage	4	0.0020	15.0
hsa04612	Antigen processing and presentation	3	0.0168	14.3
hsa05323	Rheumatoid arthritis	3	0.0234	12.0

Notes: KEGG, enrichment in Kyoto Encyclopedia of Genes and Genomes. The table shows the KEGG pathway. Enrichment analysis of DEGs with *p* < 0.05.

## Data Availability

The gene expression microarray GSE47063 (9 samples, 2 CRPS I, 2 CRPS II and 5 controls) was downloaded from the GEO database.

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
