# Peer review of "Identification of Potential Inflammation-Related Genes and Key Pathways Associated with Complex Regional Pain Syndrome"

_biomolecules, 2023, doi:10.3390/biom13050772_

Round 1

Reviewer 1 Report

Using bioinformatics analysis, the authors explored the genome transcriptional profiles of GSE47063 from 4 CRPS patients and 5 controls. Five inflammation-related genes (MMP9, PTGS2, CXCL8, OSM, TLN1) were identified to associate with complex regional pain syndrome. This study will provide helpful information to find therapeutic target for CRPS. This analysis was based on blood sample instead of symptomatic location, which will compromise the specificity of genome transcriptional profiles related to CRPS.

1.    The aim of this study is to explore the molecular mechanisms and pathogenesis of CRPS, however, there is no direct evidence to show the changes of these gene transcription are related to CRPS. It couldn’t exclude that these changes may be induced by CRPS, or parallel changes with CRPS.

2.    Lots of factors (gender, age, nerve damage, disease duration, medication, et al.) affect CRPS, the sample size (2 CRPS I, 2CRSP II) is not enough to be representative and have statistical power. Furthermore, the authors cannot rule out the medication effects on gene expression.

3.    Line 52-59, the authors summarized that increased evidence suggested that inflammation had critical role in the pathogenesis of CRPS. Is there any citation/reference?

4.    Line 64-66, Bioinformatics can provide tools for clinical diagnosis and treatment [9]. Reference 9 is basic research on a rat SNL pain model, instead of clinical diagnosis and treatment.

5.    Line 72-75, Gene Expression Omnibus (GEO) database  (http://www.ncbi.nlm.nih.gov/geo), a subdataset of the National Center of Biotechnology Information (NCBI), which includes biological expression data of many species obtained by array, SNP array and high throughput sequencing [8]. Reference 8 reported “Genome-wide expression profiling of complex regional pain syndrome”, which didn’t introduce GEO database.

6.    All data in this manuscript were from GSE47063, which looks like part of reference 8. Similar analysis was performed in this reference. However, the authors didn’t mention this paper in the discussion.

7.    Gramma, such as line 190-192, In order to further clarify the critical role of CRPS, we use the cytoHubba in Cytoscape to screen out the top hub genes MMP9, PTGS2, CXCL8, OSM, TLN1 were associated with CRPS, additionally, MYH9, ZYX and ACTN1 also play an important role in CRPS.

Reviewer 2 Report

 Introduction:

1. I suggest that perhaps the authors provide some details about implementing the bioinformatics tools and how these tools are used to identify hub genes and critical pathways for clinical diagnosis and treatment of related diseases.

2. Please provide a brief description of the current research here.

Results:

1. Most of the figures are without legends

Reviewer 3 Report

This study identified CRPS-related genes based on differential expresssion analysis and interactome database. It is good that this study seems to have show the findings of the data as it is. Moreover, its description is logical and tranditional (good).

My comments are follows. 

2.2 section) Title (2.2. Datasets analysis) is too ambiguous. Specific term, such as "Differential expression analysis" is recommended.

2.2 section) You used the representative method (GEO2R) to identify CRPS-related genes. Therefore, the GEO2R should be described in detail. For example, it should be described which method, such as t-test or moderate t-test was used and which null distribution was chosen.

2.4 section) The reference for the STRING DB should be required.

2.5 section) FDR < 0.25 for selecting pathway? What's rationale for the cut-off? Why not FDR < 0.05? Should be described in detail.

2.6 section) Nomogram is not model. It should be presented in detail which classification model and its hyper-parameter, such as logistic regression or support vector machine, was used to make the prediction model.

Figure 1 - 2)

They were same. It is recommended to integrate them into a figure.

Figure 3)

Not 37 genes, less than 37 genes. It might be error.

Figure 4, table 2)

This figure is overlapped with Table 2.

In table 2, it is recommended to present count as well as real genes overlapped.

Table 3)

Same with thoes of figure 4.

In case of this case, Table 3 is recommended to transfer into supplementary file.

Figure 5)

Specific genes overlapped is required to be described.

Figure 6)

It is better that a table including genes and their number of interaction.

Figure 7b)

What method is used to make interactions among pathways?

No result description for figure 7b.

Discussion)

It is recommended to describe the stem cell therapy to treat CRPS because mesenchymal stem cell is molecular-based therapeutic method and reduce inflammation.

Round 2

Reviewer 1 Report

 No more comments.

Reviewer 3 Report

The quality of the paper seems to be improved. I recommend editor to accept this paper.